# A Simple Field Tapping Test for Evaluating Frequency Qualities of the Lower Limb Neuromuscular System in Soccer Players: A Validity and Reliability Study

**DOI:** 10.3390/ijerph19073792

**Published:** 2022-03-23

**Authors:** Safouen Chaabouni, Rihab Methnani, Badria Al Hadabi, Majid Al Busafi, Mahfoodha Al Kitani, Khalifa Al Jadidi, Pierre Samozino, Wassim Moalla, Nabil Gmada

**Affiliations:** 1Laboratory LR19JS01 EM2S, High Institute of Sports and Physical Education, University of Sfax, Sfax 3000, Tunisia; safwanchaabouni@gmail.com (S.C.); rihabmethnani@yahoo.fr (R.M.); wassim.moalla@gmail.com (W.M.); 2Physical Education and Sport Sciences Department, College of Education, Sultan Qaboos University, Muscat 123, Oman; bhaddabi5@gmail.com (B.H.); majidb@squ.edu.om (M.B.); mkitani@squ.edu.om (M.K.); Khalifaj@squ.edu.om (K.J.); 3Humanities Research Center, Sultan Qaboos University, Muscat 123, Oman; 4Laboratoire Interuniversitaire de Biologie de la Motricité, Université Savoie Mont Blanc, EA 7424, F-73000 Chambéry, France; pierre.samozino@univ-smb.fr; 5Research Unit, “Sportive Performance and Physical Rehabilitation”, High Institute of Sports and Physical Education, Kef, University of Jendouba, Jendouba 8100, Tunisia

**Keywords:** foot tapping test, soccer players, neuromuscular function, validity, reliability

## Abstract

Over the years, the foot tapping test protocol has been proposed by scientists to identify the capabilities of the lower limb neuromuscular system in the medical context; however, to our knowledge, no studies have established its usefulness and relationship to athletic performance. The aim of the present study was to test the reliability, criterion validity and sensitivity of a new foot tapping (TAP) test, and to examine its relationship with proxies of athletic performance in soccer players. Forty voluntary soccer players of two different levels participated in this study (20 players from the national level: age: 22.6 ± 2.5 years and 20 players from regional level: 25.1 ± 3.6 years). They performed the TAP test on two separate occasions to test its relative and absolute reliability. To examine the criterion validity of the TAP test, all participants performed four types of jumps, sprint tests, agility tests, the Wingate test and the finger tapping test considered a gold standard tapping test. The sensitivity was assessed with national and regional player levels. The TAP test presented a high relative and absolute reliability with intra-class correlation coefficient ICC > 0.90, standard errors of measurement SEM < 5% and mean difference ±95% limits of agreement equal to 0.2 ± 0.8 tap·s^−1^. National level players showed a higher TAP score (*p* < 0.001; dz = 1.96, large) compared to regional players (9.68 ± 1.41 tap·s^−1^ vs. 7.28 ± 1.01 tap·s^−1^, respectively) and the value of area under curve measured by the receiver operating characteristic curve technique was 0.95 (95% CI: 0.827–0.990). The TAP test showed a significant association with the finger tapping test (r = 0.84, *p* < 0.001), whereas no correlation was seen between the TAP test and all the other physical tests measured. The TAP test could be considered a valid and reliable test to assess lower limb neuromuscular ability in soccer players.

## 1. Introduction

In recent decades, sport sciences had been largely reconstructed and developed in order to optimize performance in both training and competition. Accordingly, in most sports activities, the assessment of neuromuscular system plays a very important role to achieve higher performance [1,2]. Indeed, physical exercise is associated with various physiological changes and neuromuscular adaptation that targets the identification of performance-limiting factors [3,4].

To examine the neuromuscular responses in the upper and lower limbs many tests that allow simple and rapid assessment have been used. In soccer, for example, the practical application of different physical tests focusing on simple and rapid assessment of neuromuscular capacities should be well considered because of the financial and time implications of testing large numbers of players [5]. Read [5] emphasizes that the need for appropriate screening methods to assess neuromuscular response is important for practitioners to identify football players who may be at higher risk for injuries directly affecting their performance. In this context, the Finger Tapping Test (FGTT) proposed by Nakamura et al. [6] has been used to measure motor speed and to predict both cognitive and physical function. According to Şahin [7], the FGTT can be considered an important and effective method and tool to assess athletes in different sports branches in terms of fine motor performance, including in soccer where the evaluation of the neuromuscular responses to strength, speed and endurance and other physical qualities are essential to maximize the performance of the players [8].

In fact, the usefulness and successful application of the FGTT is very common and is often used as part of a neuropsychological examination to detect both motor and cognitive impairments [9]. In soccer, performance success has been presented as complex and highly dependent on physical, technical, tactical and neuromuscular factors [10]. Accordingly, changes of direction (COD) are an important and dominant performance factor of success in modern soccer [11]. COD is defined as a quick whole-body movement with a change of velocity or direction [12] where short and high-frequency movements are closely dependent. Despite the fact that a rapid movement is mainly caused by high muscle contraction velocity and activation, the fact of repeating it would also be related to the frequency qualities. Therefore, neuromuscular frequency qualities could be an important component of effective performance in the COD in soccer players as well as physical and technical factors such as straight sprinting, eccentric and concentric strength, power and reactive strength [11]. Indeed, the ability to repeat a single muscular action at very high frequency without introducing the influence of the velocity of muscle’s contraction is considered a key factor in soccer where rhythmic movements are essential.

In the last decade, different protocols of tapping tests have been proposed such as the surrogate finger tapping test [13] and the foot tapping test (FTT) [14,15]. However, the FTT has been widely used to identify the capacities of the neuromuscular system during small amplitude movements, repeated at a certain frequency, thus reducing the influence of the velocity of muscle’s contraction [14,15,16]. The FTT, through sensorimotor coordination tasks, appears to be a simple and easy method to quantify the motor functions of the limb due to its ability to activate and deactivate muscles at high frequency [17]. In fact, the FTT has been used to study human motor control, especially the pattern of rhythmic movements, such as bimanual coordination [18]. It has been demonstrated that a declined performance in the FTT is a predictor of a decline in gait velocity, particularly in populations with upper motor neuron problems, such as amyotrophic lateral sclerosis [19]. The FTT has also been shown to predict a decline in relation with age, older persons seem to have a declined FTT performance in comparison to younger adults, even when both populations may have the similar muscle strength and rate of force development in the tibialis anterior [20]. Although both the FGTT and FTT have shown high sensitivity and reliability and reproducibility, the FGTT seems to be more complex and involves the performance of multidimensional tasks [17].

While most research has studied the importance and the usefulness of the foot and finger tapping tests in the medical context and proving their validity in identifying a subject with declining physical function [15,21], to the best of our knowledge, no study has established the usefulness and the relationship of those tests with the sport performance where rhythmic movements are important.

On the other hand, most of the FTT protocols previously used consist of tapping the feet on the ground for a period of 10–15 s while the subject was sitting on a chair (pathological population) [14,15]. However, the effect of the subject’s body mass is eliminated which would influence performance when assessing sport physical ability and performance. Secondly, the duration of 10–15 s could induce fatigue on the neuromuscular system. Thus, we proposed a new protocol of the foot tapping test called the TAP test where the players are requested to perform the TAP in a standing position and for a shorter period (4 s) to avoid any possible neuromuscular fatigue. The aim of the present study was, therefore, to examine the reliability, criterion validity and sensitivity of this new test in soccer players from national (NL) and regional (RL) competitive levels.

## 2. Materials and Methods

### 2.1. Experimental Approach to the Problem

The reliability of the TAP was assessed using test–retest trials separated by 3 days under comparable environmental conditions whereas the criterion validity was ascertained by examining the relationships between TAP test scores and those of proxies of athletic performance of soccer players: jumping ability (drop jump [DJ], squat jump [SJ], countermovement jump [CMJ], standing long jump [SLJ]), sprint-time (5-m/20 m and 30-m), Modified Illinois change of direction test (MICODT), anaerobic leg power (Wingate) and finger tapping test (FGTT) considered a gold standard tapping test [13].

The sensitivity (discriminatory capability) of the TAP was checked according to the soccer player’s level after considering the difference that could exist between expert and non-expert sports performers in the literature (construct validity) [22], and accordingly, the TAP test needs to be able to discriminate between sports ability level.

#### 2.1.1. Subjects

A cross-sectional study was performed to estimate the correlation between two quantitative variables of interest (i.e., TAP and finger tapping test). The sample size was calculated via the following formula proposed by Serdar et al. [23]:(1)$$N=\frac{{\left(Z_{\frac{\alpha }{2}}+Z_{1-\beta }\right)}^{2}}{\frac{1}{4}\;\left[log_{e}\left(\frac{1+r}{1-r}\right)\right]}+4$$
where “*N*” was the needed number of participants; “*Z_α_*_/2_” is the normal deviates for type I error (=1.96 for 5% significance level); “*Z*_1−*β*_” was the study power (=1.64 for 95% power); and “*r*” is the correlation coefficient between the TAP and finger tapping test.

Given the pioneering character of the present study, the “*r*” was arbitrarily fixed high at 0.70. The injection of the above data in the formula gave an appraised sample size for a two-tailed alternative test of 33 participants. With an allowance for 10% of non-inclusion criteria, the corrected sample size was 33/(1.0 − 0.1) = 38 participants.

A total of forty soccer players classified in four playing positions: 4 goalkeepers, 14 defenders, 14 midfielders and 8 forwards took part in the study. These players had at least 5.3 ± 1.2 years of competitive experience in the senior team and were active in the same professional club during a minimum of two consecutive seasons. Players were recruited in a balanced way according to their playing positions from two different competition levels: (i) twenty national level soccer players (NL, age = 22.6 ± 2.5 years, body mass = 66.1 ± 8 kg, body mass index = 22.05 ± 1.63 kg/m^2^, height = 172.9 ± 8.4 cm) and (ii) twenty regional level soccer players (RL, age = 25.1 ± 3.6 years, body mass = 65.9 ± 4.9 kg, body mass index = 21.7 ± 1.58 kg/m^2^, height = 174.3 ± 3.3 cm). Players were free of injuries in the last 3 months and had regularly trained and competed in their clubs in the past 6 months. All participants provided their written informed consent after having been fully informed about the purpose of the study and the experimental protocol. The study was approved by the Ethical Committee of the university and was conducted according to the Declaration of Helsinki 1975.

#### 2.1.2. Procedures

Each participant was asked to be present on six sessions. The first session was reserved for anthropometric measurements and familiarization with all tests including the TAP and FGTT tests. The height (m) of each player was measured to the nearest 0.1 cm using a stadiometer (Holtain Ltd., Crymych, UK). Weight (kg) and body fat were determined using the Tanita MC-780 body composition analyzer (Tanita Corp., Tokyo, Japan) following the manufacturer’s instructions [24]. All anthropometric measurements were measured twice by two examiners. The second and third sessions were devoted to studying the reliability of tests. During the fourth, fifth and sixth sessions, all subjects performed the TAP, FGTT, sprint, agility, jumping tests and the Wingate test. Sessions were separated by three-day intervals and all tests were conducted randomly in the same regular indoor court. All subjects performed each test with at least 3 min of rest between all trials and 5 min between tests to ensure adequate recovery. The recovery time between the Wingate test and other tests was three days (Figure 1).

All tests were performed at the end of the regular soccer season at the same time of the day (10:00 AM–12:00 PM) in similar ambient conditions of temperature (20–23 °C) and relative humidity (60 ± 2%). Participants were instructed to wear suitable sportswear to limit possible variability within the testing procedure and were instructed to avoid intensive physical training 24 h before each testing session. They were also asked to sleep for a minimum of 7 h and avoid food and caffeine three hours before testing sessions. Furthermore, they were also instructed to maintain the same diet habits throughout the study period.

Before starting each session, the player had to warm up for 15 min consisting of 4 min of jogging, 4 min of dynamic stretching exercises, two sprints of 20 m and jumping drills, as described in the study of Dello Iacono et al. [25].
TAP test: The test was assessed with an Optojump Next (Microgate Next, Bolzano, Italy). It determines the frequency of foot tapping by calculating the time of flight/contact (one cycle) of the legs through infrared beams during 4 s (Figure 2, Appendix A). The acquisition bars were placed between the feet and connected to the Optojump Next software. The average frequency of each member was averaged on each successive right/left press to obtain the overall tapping frequency. To evaluate if TAP could cause any cardiovascular and metabolic stress, heart rate was measured twice: before and after the TAP test using (Polar V800, Finland). Blood lactate was also measured at rest before performing the TAP test and at the third minute following test using Lactate Pro Analyzer (Arkray, Tokyo, Japan).The finger tapping test: The FGTT procedure used in this study was the one described in the study of Austin et al. [13]. Three trials have been performed each for both the dominant and non-dominant hands and scores were measured with a smartphone application (HLTapper V.1.0), validated by Lee et al. [26]. This application is composed of two rectangles of 30 by 45 mm, separated by 15 mm. Smartphone timed tapping test subjects were asked to alternately tap each side of the rectangles using an index finger at their fastest speed for ten seconds without moving the rest of their hand or arm. The FGTT score is reported as the best number of taps recorded during the 6 trials (both dominant and non-dominant hand) then the tapping score was calculated as FGTT = Score/10 s.Sprint test: Sprint performance was evaluated at 5–20 and 30 m intervals through an electronic timing system (Brower timing system, CO, Draper, UT, USA). Players started in a standing start position 0.3 m just before the first photocells gate, placed at 0.75 m above the ground. The best time performance from 2 trials with a 3 min rest in-between was chosen for analysis.Agility test: The MICODT was used to evaluate the agility performance, where the protocol is the same as the Illinois CODS test with the only difference being in the total distance, inter-cones distance, and the number of the cones as described by Hachana et al. [27] and using an electronic timing system (Brower timing system, CO, Draper, UT, USA).Jump Tests: For SJ, players have started from a stationary semi-squatted position and performed a vertical jump at maximal effort. Then, players performed 2 types of CMJ, both where the jump starts in the standing position and the subject performs a downward countermovement (flexion of the lower limbs) immediately followed by a rapid full extension of the lower limbs. The difference between the 2 types of CMJ was that, in the first type, players should maintain their arms akimbo, but in the second type of CMJ, the players were asked to keep their arms in a neutral position (free hands). The jump height was recorded using an Optojump device (Microgate Next, Bolzano, Italy). The eccentric utilization ratio (EUR) has also been calculated, which is defined as the ratio of the CMJ (arms akimbo) to SJ performance [28].

For SLJ, participants standing with their feet at approximately shoulders’ width were requested to jump from the departure line to the furthest distance possible. The jump distance was measured then from the takeoff line (toe line) to the point of heel placement at landing.

Regarding the Drop jump (DJ), with their hands on their waist players were asked to step off the platform (40 cm height from the ground), they were requested to jump off as fast as possible and then to jump as high as possible making sure that the knees and ankles were completely extended when leaving the ground and in a similarly extended position when landing. The reactive strength index (RSI), defined as the quotient of the jump height and contact time, was calculated [29].

All jumping tests were repeated 3 times with a 3 min rest in-between and the highest performance was considered for further analysis.
Anaerobic power: The Wingate anaerobic test (WAnT) was performed on a cycle ergometer Monark (Monark 894E, Stockhom, Sweden). The test was preceded by a warm-up of 5 min in approximately 100 rpm, with two sprints of approximately 6 s every minute, followed by a 2 min rest interval before the start of the test. Each participant had to exert maximal effort for 30 s against a braking force that was determined by the product of body mass in kg by 0.075 [30]. Two indices of the WAnT were measured and evaluated: (a) the peak power (Ppeak) and (b) the mean power (Pmean), both expressed in Watts per kilogram of body mass.

### 2.2. Statistical Analyses

Data were presented as means ± standard deviation. All data analyses were analyzed using IBM SPSS Statistics version 23.0 software (SPSS Inc., Chicago, IL, USA). Normal distribution was checked by Shapiro–Wilk test. A dependent samples *t*-test was applied to detect any learning effect or systematic bias between test and retest scores.

Relative and absolute reliability of the TAP and FGTT tests was evaluated by ICC and the standard error of measurement (SEM) expressed as coefficient of variation (CV), respectively. Bland and Altman method was also used to evaluate TAP absolute reliability [31]. The usefulness of the TAP was checked by comparing the smallest worthwhile change (SWC) and the SEM. When multiplying the between-players SD by 0.2, the SWC0.2 could indicate then the typical small effect or by multiplying by 0.6 the SWC0.6 could show an alternative moderate effect. According to Hopkins [32], when the SEM was less, similar, or higher than the SWC, its ability to detect a small or moderate change was rated as “good”, “OK” or “marginal”, respectively.

The minimal detectable change at 95% of confidence interval (MDC95%) was calculated to determine the magnitude of change that must be observed before the change would be considered to exceed the measurement error and variability at the 95% confidence level. The MDC95% was calculated based on the standard error of measurement by using the following formula:(2)$$MDC95\%=1.96\times \sqrt{2}\times SEM$$
where 1.96 is the z score of the 95% CI from a normal distribution, and √2 is used to account for the error of scores from measurements at 2 points in time [33].

Pearson’s correlation coefficient was used to establish the association between the TAP and the other physical tests measured. According to Batterham and Hopkins [34], the effect was qualitatively considered as follows: trivial if r < 0.1, small if 0.1 < r < 0.3, moderate if 0.3 < r < 0.5, large if 0.5 < r <0.7, very large if 0.7 < r < 0.9, nearly perfect if r > 0.9, and perfect if r = 1. The coefficient of determination (R^2^) was calculated then to evaluate the level of common variance between the TAP test and the other physical performance measured.

After checking distribution normality (Shapiro–Wilk test), an unpaired Student *t*-test was used in order to check the difference between levels of NL and RL players in TAP score. The effect size (Cohen effect size, dz) was also calculated for each output [35]. The criteria to interpret the magnitude of the dz were as follows, small if (0.00 ≤ dz ≤ 0.49), medium if (0.50 ≤ dz ≤ 0.79) and large if (dz ≥ 0.80).

The sensitivity of the TAP test (construct validity) was evaluated through the receiving operator characteristic (ROC) curve analysis. According to Delacour et al. [36], the ROC curve is a method for studying the clinical efficacy of a test. Indeed, the comparison of the areas under the curve makes it possible to appreciate and classify the diagnostic performances of several test levels, better than the simple study of the coupling sensitivity specificity. ROC curve is a graphic representation of the relation existing between the sensibility and the specificity of a test, and according to Deyo and Centor [37], if the area under the ROC curve (AUC) > 0.70, then it is commonly considered to indicate the good discriminant validity of the test. Significance for all statistical tests was set at *p* < 0.05.

## 3. Results

The Anthropometric, physical and physiological characteristics of subjects are summarized in Table 1.

Statistical analysis showed that all data are normally distributed (*p* > 0.05). There was no significant difference between the TAP test and retest scores measured (8.35 ± 1.74 tap·s^−1^ vs. 8.33 ± 1.71 tap·s^−1^; *p* = 0.71).

The reliability of the TAP indicated that the ICC value was higher than 0.90, the SEM expressed as CV was less than 5% and was lower than both SWC0.2 and SWC0.6 (Table 2).

The FGTT showed an ICC value higher than 0.90 (Table 3). A high level of concordance between scores of the TAP test–retest was verified by Bland and Altman plots (Figure 3).

The residual errors between TAP scores on test and retest were normally distributed (*p* = 0.417). The mean difference (bias) ± the 95% limits of agreement (LOA) was 0.2 ± 0.8 tap·s^−1^. The MDC95% showed a value of 0.8 tap·s^−1^.

Regarding the difference between the two soccer player groups’ level, the results showed that NL players presented a higher TAP score (*p* < 0.001; dz = 1.96, large) in comparison with RL players (9.68 ± 1.41 tap·s^−1^ vs. 7.28 ± 1.01 tap·s^−1^, respectively). The average difference in the blood lactate level (Δ[Lac]) and heart rate (ΔHR) measured before and after performing the TAP test was low for both groups (2.1 ± 1.8 mmol·L^−1^; 36.2 ± 13.3 bpm and 2.5 ± 1.4 mmol·L^−1^; 32.8 ± 11.3 bpm, respectively, for NL and RL groups). The value of area under the curve (AUC) measured was 0.95 (95% CI: 0.827–0.99, *p* = 0.03). The resulting cut-off for the TAP score was <8.26 tap·s^−1^ (Figure 4).

## 4. Discussion

The present study aimed to examine the reliability, criterion validity and sensitivity of the TAP test in soccer players. The main finding indicated that the TAP test is a reliable, valid and sensitive test. Moreover, the TAP test showed no significant relationship with all the physical tests measured, which supports its specificity to assess another ability than those targeted by these tests.

The physiological characteristics evaluated in this study showed that the average difference in the blood lactate level (Δ[Lac]) and heart rate (ΔHR) measured before and after performing the TAP test were low, indicating that the TAP test did not induce significant cardiovascular and/or metabolic stress.

The TAP test showed a high relative and absolute reliability (ICC = 0.98 and SEM = 3.5%, respectively) and lower SEM values in comparison with those of SWC0.2 and SWC0.6 meaning good ability of TAP test to detect significant performance changes. The data of reliability of the TAP test evaluated by calculating the 95% LOA described by Bland and Altman [31] showed that both bias and random error were low, resulting in good absolute reliability. In fact, if a player scored 8 tap·s^−1^ on their first trial of the TAP test, it suggests that on the second trial they could perform as many as 8.8 tap·s−1 (8.0 + 0.8 tap·s^−1^), or as few as 7.2 tap·s^−1^ (8 − 0.8 tap·s^−1^). Those results are in accordance with Hinman’s [15] study which has examined the test–retest reliability of the foot tap test on the dominant foot among younger and older adults. In this study, despite the difference in the protocol, where the subjects were seated on a chair with their knees in approximately 90 degrees of flexion and were asked to tap their dominant foot as many times as possible for 10 s, the results showed good relative reliability (ICC = 0.79, *p* < 0.001) as well as the LOA analysis which showed a good absolute reliability between the trials. This result corroborates those of Pribble et al. [38] who pointed out that the foot taping test presented high test–retest reliability when evaluating the foot tapping test in a healthy population. Unfortunately, to best of our knowledge no other study has studied the interest of the foot tapping test with healthy subjects or in sports filed and the sole data found in literature have been done in the medical context with pathological subjects.

Our data were in agreement with Raghavendran [17], who reported that the foot tapping test presented a high inter-evaluator reliability and has consistent reproducibility and sensitivity in detecting corticospinal tract involvement, and with Numasawa et al. [16], who demonstrated a good reliability of the foot tapping performance test evaluated in patients with cervical compressive myelopathy. However, these authors did not find an acceptable reliability of foot tapping in a normal subject.

The minimal detectable change (MDC95) 95% was calculated to show the minimum value of change from which any difference should be related to an improvement rather than measurement errors. The MDC95% values calculated in the present study were 0.8 tap·s^−1^. These results indicated that a change in TAP score over this value could be considered “significant” and reflect a real performance improvement in soccer players.

In the literature, the foot tapping test has been used mostly in medical studies evaluating movement speeds in patients with neuromuscular disorders [16,19]. The foot tapping test was always considered a simple physical test that allows the rapid assessment of neuromuscular function [14]. This could explain why the TAP test does not correlate with all the other physical performance measured except the finger tapping test. In fact, except for the motor neuromuscular ability, the TAP test seems to be not affected by the players’ physical ability such as muscle strength, muscle elasticity, power, or even velocity. It could be considered a test that measures only the quality of the neuromuscular system and was not adequate to the physical assessment and sport performance. This finding is in accordance with the results of Eduard et al. [39] who reported that the Foot tapping tests are identified as completely independent of strength qualities and therefore present only a valuable addition to strength-related performance testing, thus supporting the idea that TAP test is not a good indicator of the physical fitness. In the same context Voss G et al. [40] reported that in high performance sports, tapping tests examine cyclical neuromuscular performance and are used for talent identification. Likewise, according to Zemková et al. [2] the evaluation of physical abilities such as speed can be only supplemented by testing the frequency of movements of the upper and lower limbs through the tapping tests which can be done while standing or sitting. Indeed, ŞAHİN et al. [7] reported that tapping test is also accepted as an important method for evaluating the athletes in different sports branches in terms of fine motor skill performance including in soccer and where the assessments of neuromuscular responses are important [5] and deeper knowledge should be gained by objectively measuring neuromuscular performance capabilities [4].

Recently, Enoki et al. [14] found that the 10 s foot tapping speed was significantly correlated with the 30 m walking speed evaluated by the simple walking test, but it’s important to highlight that the study was done with patients with cervical compression myelopathy where the 30 m walking speed was considered a gold standard test to quantify the slowness of voluntary leg movements in this pathological condition. In contrast, Numasawa et al. [16] reported that the value of foot tapping does not always influence the ability to walk due to many factors such as muscle weakness which have been shown to affect the locomotor function in cervical myelopathy. Besides, the authors pointed out that the FTT cannot accurately assess subjects who have suffered from muscle weakness.

The significant correlation (70% shared variance) between the TAP and FGTT tests confirm the criterion validity of the TAP. The high association between both tapping tests supports the results of Tanigawa et al. [41] who have shown that the finger and foot tapping tests are related to each other and considered to be functional screens for individuals’ motor functions. Furthermore, Tanigawa et al. [41] found a higher sensitivity to detect neuromuscular disorder when both tests were combined confirming the useful association of the two tests.

Regarding the sensitivity of the TAP test (i.e., construct validity) which was evaluated by measuring its ability to detect the difference in performance that may exist between NL and RL soccer players, NL group showed significantly higher TAP test scores comparatively to RL group. These results were confirmed through the analysis of the AUC derived from the ROC curves. This statistical test is more appropriate to examine the discriminant ability because a difference, although significant, does not necessarily imply that the variable is able to discriminate. Indeed, the ROC curve cutoff value relative to TAP test was <8.26 tap·s^−1^ indicating that an equal or higher TAP score than 8.26 tap·s^−1^ is considered a high TAP score performance. This result seems to be interesting and should be taken into consideration by coaches and scientists, when they established a training sports program to increase the TAP performance in soccer players.

Finally, to our knowledge this is the first study that has investigated the test–retest reliability and validity of foot tapping test in soccer players (sports population) and not in patients (medical context). Accordingly, the TAP could be considered as a simple quantitative method to assess the neuromuscular responses in soccer players. The current results did not show any relationship between the TAP score and physical ability tested. Nevertheless, the only correlation was observed with the finger tapping score. Therefore, further research is needed to firstly, evaluate the relationship between the TAP and other physical qualities such as reaction time, coordination and balance performance, which we did not evaluate in our present study, as well as using a different agility test, which may have different results since the MICOD test could be not fit for such a category of soccer group’s level. Secondly, future studies aiming to investigate whether TAP could be applied as post-activation potentiation (PAP) to enhance sport performance will be pertinent since PAP has always been related to the neuromuscular function.

## 5. Conclusions

The main conclusion of this study indicates that the TAP test is a valid and reliable test for assessing frequency of the lower limb neuromuscular system in soccer. In addition, the TAP test is a simple filed test that can be used by coaches to evaluate the ability to repeat a single muscular action at very high frequency without introducing the influence of the velocity of muscle’s contraction. In fact, in addition to the classic physical qualities trained during practice sessions in order to improve the physical condition of athletes such as strength, speed, power and endurance, the TAP should also be used as a specific exercise to improve the frequency qualities of the neuromuscular system considered as a determining factor in sport performance. For that, coaches and/or fitness trainers are strongly recommended to integrate the TAP into their assessment and training strategies in the future, and specific work based on the repetition of TAP should be planned into the weekly training program of the players to increase the lower limb neuromuscular responses by increasing the TAP score performance.

## Figures and Tables

**Figure 1 ijerph-19-03792-f001:**
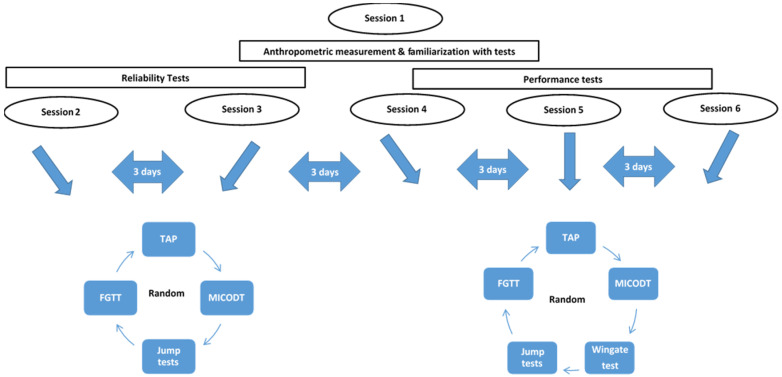
Schematic diagram of experiment.

**Figure 2 ijerph-19-03792-f002:**
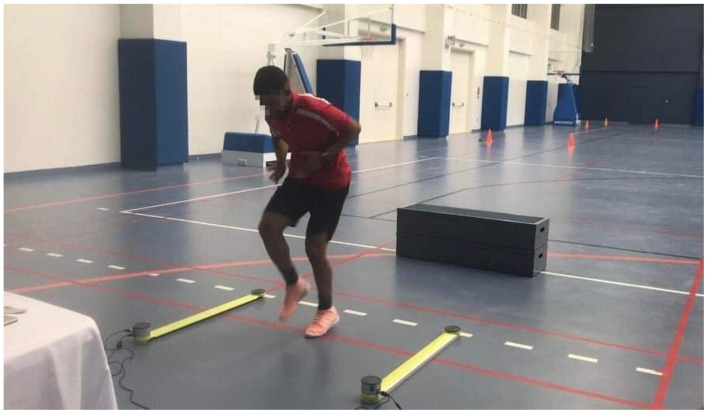
TAP test. In standing up position, the players are asked to tape the feet as quickly as possible on the ground with two alternating limbs for 4 s using Optojump Next (Microgate Next, Bolzano, Italy).

**Figure 3 ijerph-19-03792-f003:**
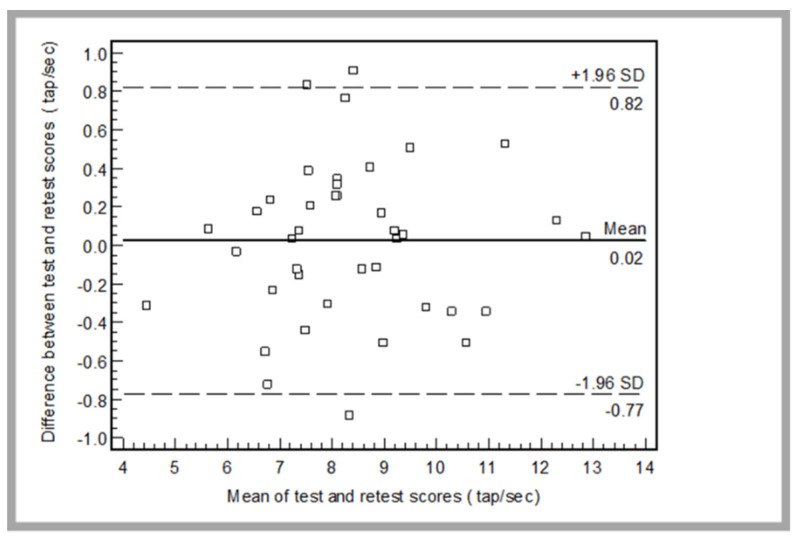
Bland and Altman plots with limit of agreement (dashed line) of test–retest of the TAP. The differences between test–retest scores (TAP score2–TAP score1) plotted against their mean (dot line) for each subject.

**Figure 4 ijerph-19-03792-f004:**
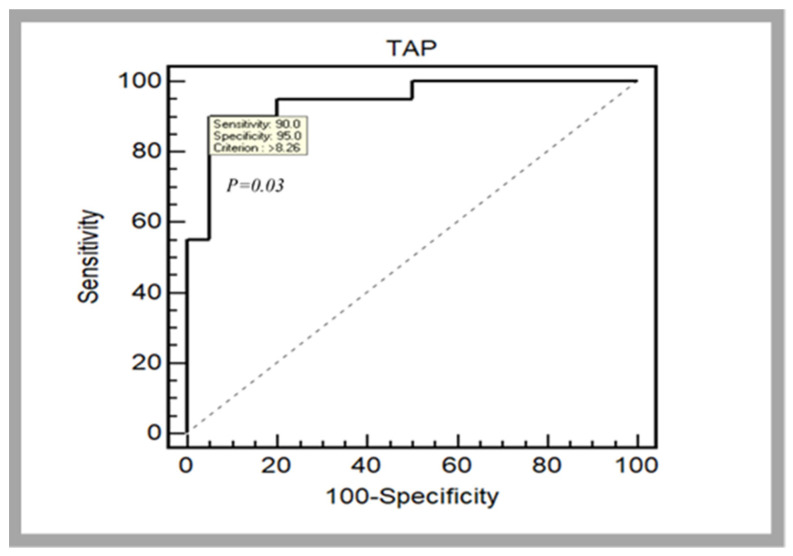
Receiver operating characteristics (ROC) curve for the TAP test between national and regional group level indicating that an equal or higher TAP score than 8.26 tap·s^−1^ is considered a high TAP score performance.

**Table 1 ijerph-19-03792-t001:** Anthropometric and physiological parameters in soccer player groups.

Parameters	N	Age (Years)	Height (cm)	Body Mass (kg)	BMI (kg/m^2^)	Body Fat %	ΔHR (bpm)	Δ[Lac] (mmol/L)	TAP (tap·s^−1^)	TAP Dz
National level	20	22.6 ± 2.5	172.9 ± 8.4	66.1 ± 8	22.05 ± 1.63	10.9 ± 1.2	36.2 ± 13.3	2.1 ± 1.8	9.68 ± 1.41 ******	1.96
Regional level	20	25.1 ± 3.6	174.3 ± 3.3	65.9 ± 4.9	21.69 ± 1.58	11.3 ± 1.4	32.8 ± 11.3	2.5 ± 1.4	7.28 ± 1.01	

******: *p* < 0.001. BMI: Body mass index; HR: Heart rate; Δ[Lac]: Average difference in the blood lactate concentration; TAP: Foot tapping test; Dz TAP: Cohen effect size.

**Table 2 ijerph-19-03792-t002:** Reliability results of TAP.

Parameters	*Trial 1*	*Trial 2*	*p* (*t* Test)	ICC (95% CI)	SEM (tap·s^−1^)	SEM (%)	SWC0.2 (tap·s^−1^)	SWC0.6 (tap·s^−1^)	MDC95% (tap·s^−1^)
**TAP** (**tap·s^−1^**)	8.35 ± 1.74	8.33 ± 1.71	0.71	0.98 (0.97–0.99)	0.28	3.52	0.34	1.03	0.8

TAP: Foot Tapping Test; ICC: Intra Class Correlation; CI: Confidence Interval; SEM: Standard Error of Measurement; SWC: Smallest worthwhile change; MDC95%: Minimal detectable change.

**Table 3 ijerph-19-03792-t003:** Mean ± SD of players performances, Intraclass Correlation between trials 1 and 2 (ICC), correlation coefficient (r) between TAP test and other physical qualities and associated probability values (*p*).

Test	Trial 1 (Mean ± *SD)*	Trial 2 (Mean ± *SD)*	ICC	Best Performance (Mean ± *SD)*	R (95%CI)	*p*
TAP (tap·s^−1^)	8.35 ± 1.74	8.33 ± 1.71	0.98	8.48 ± 1.71	-	-
FGTT (tap·s^−1^)	7.77 ± 1.19	7.81 ± 1.07	0.96	7.95 ± 1.1	0.84 (0.77 to 0.89)	<0.001
5 m sprint(s)	1.21 ± 0.12	1.22 ± 0.11	0.85	1.19 ± 0.17	−0.16 (−0.44 to 0.14)	0.32
20 m sprint(s)	3.26 ± 0.14	3.24 ± 0.15	0.86	3.21 ± 0.15	−0.17 (−0.45 to −0.12)	0.28
30 m sprint(s)	4.33 ± 0.19	4.30 ± 0.19	0.91	4.28 ± 0.19	−0.12 (−0.42 to 0.25)	0.44
MICOD(s)	10.66 ± 0.51	10.51 ± 0.45	0.93	10.48 ± 0.46	−0.1 (−0.4 to 0.22)	0.52
SJ (cm)	37.08 ± 4.53	35.44 ± 4.71	0.97	37.16 ± 4.62	0.02 (−0.31 to 0.35)	0.85
CMJ arms akimbo (cm)	39.28 ± 5.35	39.19 ± 5.55	0.96	39.98 ± 5.45	−0.04 (−0.32 to 0.19)	0.78
CMJ free arms (cm)	44.82 ± 6.2	45.16 ± 6.71	0.95	45.98 ± 6.67	0.09 (−0.17 to 0.35)	0.58
SLJ (m)	2.16 ± 0.21	2.17 ± 0.22	0.94	2.2 ± 0.21	0.05 (−0.33 to 0.42)	0.75
VDJ (cm)	33.91 ± 4.9	34.75 ± 5.57	0.95	35.31 ± 5.43	0.24 (0.03 to 0.43)	0.12
RSI	94.20 ± 26.29	97.07 ± 31.71	0.85	96.52 ± 32.2	−0.12 (−0.35 to 0.1)	0.45
Ppeak (watts)	821.62 ± 144.6	-	-	821.62 ± 144.6	−0.15 (−0.46 to 0.2)	0.34
Pmean (watts)	581.9 ± 99.4	-	-	581.9 ± 99.4	−0.08 (−0.42 to 0.34)	0.59
RPpeak (watts/kg)	12.19 ± 1.41	-	-	12.19 ± 1.41	−0.04 (−0.34 to 0.25)	0.79
RPmean (watts/kg)	8.62 ± 0.77	-	-	8.62 ± 0.77	0.08 (−0.24 to 0.39)	0.62
EUR	1.08 ± 0.08	-	-	1.08 ± 0.08	−0.12 (−0.38 to 015)	0.45

5–20–30 m sprint: Sprint time test; MICODT: Agility test; RPpeak: Relative peak power; RPmean: Relative mean power; SJ: Squat jump; CMJ: Countermovement jump (arms akimbo), Countermovement jump (free arms); EUR: Eccentric utilization ratio; SLJ: Standing long jump; VDJ: Vertical drop jump; RSI: Reactive strength index; TAP: Foot Taping Test; FGTT: Finger Tapping test; CI: Confidence Interval.

## Data Availability

The datasets generated for this study are available on request to the corresponding author.

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
