# Peer review of "A Simple Field Tapping Test for Evaluating Frequency Qualities of the Lower Limb Neuromuscular System in Soccer Players: A Validity and Reliability Study"

_ijerph, 2022, doi:10.3390/ijerph19073792_

Round 1

Reviewer 1 Report

Title: A simple field tapping test for evaluating frequency qualities of the lower limb neuromuscular system: Validity and reliability in soccer players

The work aims to explore the reliability, criterion validity and sensitivity of a new foot-tapping (TAP) test, and to examine its relationship with proxies of athletic performance in soccer players, which is an interesting issue related with player’s performance. The methodology, tools and statistical approach are adequate. The work is clearly presented and the document is generally well written, however, some minor revisions are suggested.

TITLE could be improved in order to better reflect the aim of the work, suggestion: “A simple field tapping test for evaluating frequency qualities of the lower limb neuromuscular system in soccer players: Validity and reliability study”.

ABSTRACT SECTION

Lines 32-33- Conclusion should focus on the TAP test information. Suggestion: “The TAP test could be considered as a valid and reliable test to assess lower limb neuromuscular ability in soccer players.

SECTION 1. INTRODUCTION

The authors should consider including the following references:

doi:10.1519/JSC.0000000000002069

https://doi.org/10.3390/ijerph17239147

https://doi.org/10.3389/fphys.2018.00264

SECTION 2. MATERIALS AND METHODS

Line 161. Figure 1 The caption should include a a short explanatory title and description of the instrument used to record the frequency of foot tapping.

Line 249. Authors should mention the validation of the assumptions for the application of the selected test

Line 261. p (lowercase letter) should be italic

SECTION 3. RESULTS

Line 266. p (lowercase letter) should be italic

Figure 2 and 3 should have a short explanatory title and captions should describe in detail what is depicted.

SECTION 4. DISCUSSION

The authors should consider including in the discussion the following references:

doi:10.1519/JSC.0000000000002069

https://doi.org/10.3390/ijerph17239147

https://doi.org/10.3389/fphys.2018.00264

SECTION 5. CONCLUSION

Line 403-404. Authors should mention that the test applied evaluates the lower limbs. Suggestion: The main conclusion of this study indicates that TAP test is a valid and reliable test for assessing frequency of the lower limbs neuromuscular system in soccer.

Reviewer 2 Report

In this study the authors aimed to test the reliability, criterion validity and sensitivity of a new foot-tapping (TAP) test, and to examine its relationship with proxies of athletic performance in soccer players.

Although the study has the potentiality of being shared with the scientific community, I believe that the manuscript would benefit from a minor revision with the attempt to better support their experimental setting.

  1. Abstract: they should start with a first paragraph describing the background. 
  2. Methods section: More information should be provided about the participants’ characteristics. Moreover, they should describe an anthropometric measurements protocol.
  3. Discussion: which were the limits and strength of the study?
  4. Conclusion: I would like to see more of the practical implications. Based on the analyzed variables, how the authors intend to use their findings?

Kind regards

Reviewer 3 Report

First, I would like to recognize the authors for the data they collected to examine the reliability, criterion validity and sensitivity of a new foot-tapping (TAP) test and its relationship with proxies of athletic performance in soccer players. Considering that this new TAP test could be considered a valid and reliable test to assess neuromuscular ability in soccer players, this study's results are important. Indeed, no previous studies have examined the usefulness and the relationship of those tests with sports performance where rhythmic movements are essential. Furthermore, using a TAP in a standing position is new and innovative and makes perfect sense to be used in an athletic population rather than the sitting test, which is commonly used in pathological cases.

The title is clear, concise, and informative.

The abstract is clear and includes the objectives, design, methods, variables considered, main results and most relevant conclusion.

The introduction is clear and follows a logical sequence while all the relevant scientific support is provided. Furthermore, the objectives are clearly set out!

Line 56: finger tapping test, foot-tapping test. I would use "and foot-tapping test" as I was expecting to see another test after the foot-tapping test.

In the methods section, you stated that half players were national level players while the other half were regional level players. Out of the 40 total, you stated that you included four goalkeepers, 14 defenders, 14 midfielders and eight forwards. Did you choose half from each level? (2 goalkeepers from regional and two from national level?). I confirmed that when you stated that they were chosen in a "balanced way", I needed to make sure that I understood that correctly.

Line 128: no need for a space before "players".

The methodology and techniques are adequate to reach the objectives of the study. Moreover, the data, materials, sources are clearly presented so that someone can replicate the study.

The results are clearly presented and correspond to the data obtained. Furthermore, the results provide relevant information in terms of the study's objectives. The figures and tables are appropriate and sufficient.

The discussion is clearly presented and supported. In line 389, I would not say it seems to be the first. I would say to our knowledge this is the first study that… (minor comment).

Reviewer 4 Report

The authors aimed to investigate the reliability, criterion validity, and sensitivity of a new test (TAP test) in soccer players from national (NL) and regional (RL) competitive levels. The current study has merit but major concerns must be addressed. 

Introduction
The introduction is very unspecific for the study's proposal. In the first paragraph, the authors talk about the clinical application of the TAP test. However, the current study is conducted with healthy soccer players. After that, the sportive context is introduced but a generic description was present. The rationale of TAP application in soccer players was not considered. "FIT" was not described. Thus, a new rationale should be presented by the authors. The TAP test should be better introduced as a new proposal of fitness assessment in the sports field.

Methods

The aims must be not presented again in the methods. A figure describing the study's design should be helpful. A video of the TAP test application should be proposed as supplementary data. 

Results

1) Tables labels must describe the tables' content.

2)The tables should be self-explaining. 

3)The tables must have a title and a label.

4)Sometimes the authors used a label as a title. 

Discussion

This section has the same problem as the introduction. Since the TAP test was proposed in the sports field the authors should discuss the data from other studies that employed this test (or similar tests) in the same context. Why the clinical application have been discussed ? Please clarify how is the objective of the current study.

Round 2

Reviewer 4 Report

The manuscript is improved. All my concerns were addressed.